# Incomplete Multi-View Multi-Label Classification via Shared Codebook and Fused-Teacher Self-Distillation

**Xu Yan[1], Jun Yin[1]\*, Shiliang Sun[2]\*, Minghua Wan[1]**
[1]College of Information Engineering, Shanghai Maritime University
[2]School of Automation and Intelligent Sensing, Shanghai Jiao Tong University
`yanxu@stu.shmtu.edu.cn`, `{junyin,mhwan}@shmtu.edu.cn`
`shiliangsun@gmail.com`

## Abstract

Although multi-view multi-label learning has been extensively studied, research on the dual-missing scenario, where both views and labels are incomplete, remains largely unexplored. Existing methods mainly rely on contrastive learning or information bottleneck theory to learn consistent representations under missing-view conditions, but loss-based alignment without explicit structural constraints limits the ability to capture stable and discriminative shared semantics. To address this issue, we introduce a more structured mechanism for consistent representation learning: we learn discrete consistent representations through a multi-view shared codebook and cross-view reconstruction, which naturally align different views within the limited shared codebook embeddings and reduce feature redundancy. At the decision level, we design a weight estimation method that evaluates the ability of each view to preserve label correlation structures, assigning weights accordingly to enhance the quality of the fused prediction. In addition, we introduce a fused-teacher self-distillation framework, where the fused prediction guides the training of view-specific classifiers and feeds the global knowledge back into the single-view branches, thereby enhancing the generalization ability of the model under missing-label conditions. The effectiveness of our proposed method is thoroughly demonstrated through extensive comparative experiments with advanced methods on five benchmark datasets. Code is available at `https://github.com/xuy11/SCSD`.

## 1 Introduction

Multi-view data are very common in the real world (Zhao et al., 2017), where a single sample is often described by multiple representations from different modalities or various feature extraction methods, such as RGB/HSV/GIST for images, audio-visual synchronization for videos, content/behavior/social views in recommender systems, and multi-omics data in bioinformatics (Yan et al., 2021). The goal of multi-view learning is to exploit the consistency and complementarity among views to improve the quality of representations and the performance of downstream tasks such as classification. It has already become a fundamental technique in numerous real-world applications (Yu et al., 2025).

Similarly, many tasks naturally fall into the multi-label setting, where a single sample is often associated with multiple labels, such as in image classification and multi-topic text classification (Hang & Zhang, 2021). Multi-label classification can improve prediction performance by exploiting label correlations (Chen et al., 2019). If such correlations are effectively modeled and utilized, they not only alleviate the negative impact of label sparsity but also enhance prediction accuracy and robustness under limited annotation conditions.

However, the ideal assumption of complete multi-view data with fully observed multi-label annotations is rarely satisfied in practice (Wen et al., 2023). On the one hand, incomplete multi-view

---

\*Corresponding author.

data are very common (Yin & Sun, 2021). During multi-view data collection, sensor failures, occlusions, or cross-domain restrictions (e.g., privacy and authorization constraints) often render certain views unavailable during training or inference. On the other hand, missing multi-label data are also prevalent (Chen et al., 2020). This is mainly due to the high cost of fine-grained annotation and the limited attention of annotators, which often result in only partial labels being observed for some samples. Treating missing labels as negative instances in a naive way further aggravates the class imbalance problem and introduces bias (Ridnik et al., 2021).

A more challenging scenario arises when both multi-view and multi-label data are missing simultaneously, forming the dual-missing situation (Liu et al., 2023b). Firstly, missing multi-view data affect the learning of consistency and complementarity across views, increasing the uncertainty of representation learning. Secondly, missing multi-label data compromise the modeling of label correlations and the completeness of supervisory signals. When both types of missingness occur at the same time, methods designed to handle only one type of missingness often fail to be effective (Tan et al., 2018).

In response to this challenge, systematic research on the problem of Incomplete Multi-View Multi-Label Classification (IMVMLC) has significant practical and theoretical value. This study mainly focuses on two existing technical directions. The first is multi-view consistency representation learning. Representative works include DICNet (Liu et al., 2023b), which is based on contrastive learning and enforces representation consistency by constructing positive pairs across different views, and SIP (Liu et al., 2024c), which follows the information bottleneck principle to maximize shared information by preserving effective features while minimizing non-shared information. The second direction is multi-view fusion strategies, which include early fusion, intermediate fusion, and late fusion. Various representative methods explore different fusion paradigms. For example, AIMNet (Liu et al., 2024a) adopts average fusion to obtain robust but relatively "smoothed" predictions. LMVCAT (Liu et al., 2023c) introduces learnable weights to adaptively allocate the contribution of each view feature, thereby improving discriminability. RANK (Liu et al., 2025) employs a view-quality-aware subnetwork to explicitly leverage multi-view complementarity, enabling the classification network to learn reliable cross-view fused representations.

However, these methods face certain limitations. In learning multi-view consistency representations, they often rely on loss-based constraints (e.g., contrastive learning) or regularization techniques that minimize non-shared information across views. When views are missing, such strategies easily lead to under-representation or over-regularization, which limit the generalization ability of the model. Moreover, most existing fusion strategies overlook the structural information implied by label correlations, and many learnable-weight-based or quality-discriminator-based fusion approaches introduce additional training costs.

To address these issues, we propose a method, Incomplete Multi-View Multi-Label Classification via **S**hared **C**odebook and Fused-Teacher **S**elf-**D**istillation (SCSD), as shown in Figure 1. First, for consistency representation, we introduce a shared codebook and cross-view reconstruction mechanism. The shared discrete codebook captures cross-view common semantics, while cross-view reconstruction further enhances the consistency of the discrete representations. The limited multi-view shared codebook embeddings reduce feature redundancy and enhance the generalization ability of the representations. Second, for decision fusion, we design a label-correlation-oriented fusion strategy. This strategy assigns different weights to each view by estimating the ability of each view prediction to preserve the original label correlation structure, thereby reducing the impact of low-quality views. Finally, for the training paradigm, we adopt fused-teacher self-distillation: the fused prediction serves as the teacher signal to guide the learning of each view-specific classifier. In this way, the global knowledge integrated across views is fed back into the single-view branches, improving consistency, robustness, and generalization during both training and inference. The main contributions of this paper are summarized as follows:

- We propose a novel framework for incomplete multi-view multi-label classification based on a shared codebook and fused-teacher self-distillation. The framework handles arbitrary missing scenarios and achieves leading performance on multiple datasets, surpassing many advanced methods.

- We propose to learn discrete consistent representations through a multi-view shared codebook, which quantizes continuous features into a limited set of codebook embeddings. This design produces more compact representations and effectively reduces redundant informa-

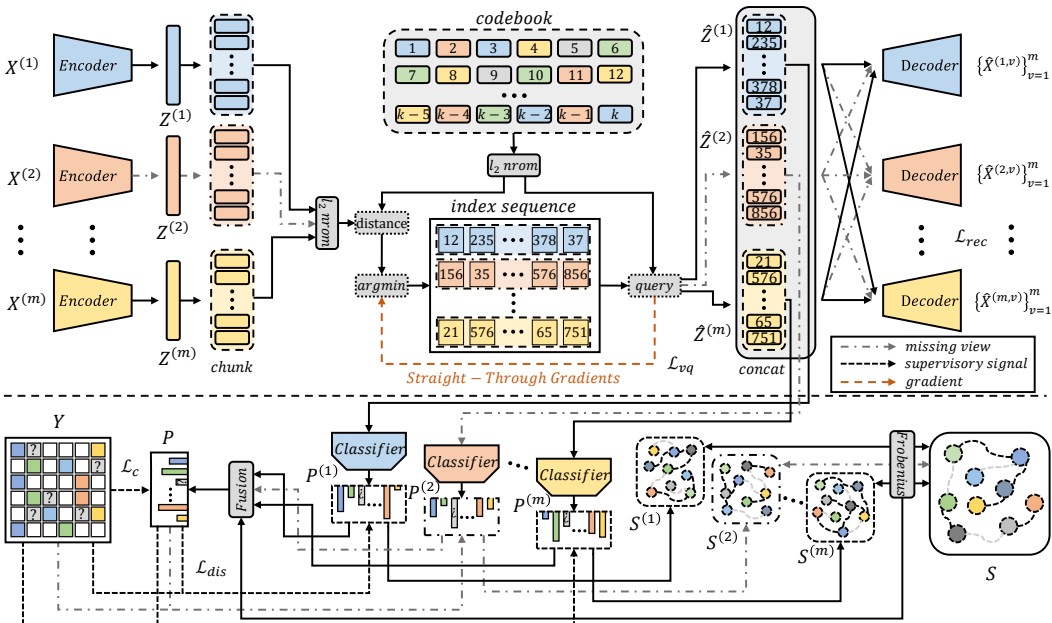

Figure 1: The main framework of SCSD. The upper part represents the framework of multi-view consistent discrete representation learning, while the lower part represents the framework of multi-view prediction fusion and self-distillation.

tion. At the same time, the features of different views can naturally align in this shared codebook embedding space, which enhances the consistency of multi-view representations.

- We propose a weighted fusion method that assigns weights according to each view's ability to preserve label correlation structures in its predictions. This method does not rely on additional external networks or learnable weights and fully exploits the structural information inherent in the supervision signals.

- We introduce a fused-teacher self-distillation framework for multi-view predictions, in which the knowledge of all views is fed back to each view branch through a self-distillation loss, thereby improving the generalization ability of the model.

## 2 METHOD

### 2.1 PROBLEM DEFINITION

In this section, we define the problem and introduce the notations. We consider a multi-view dataset $\{X^{(v)}\}_{v=1}^m$, where $m$ denotes the number of views, and $X^{(v)} \in \mathbb{R}^{n \times d_v}$, with $d_v$ representing the original feature dimension of the $v$-th view and $n$ representing the number of samples. We define a label matrix $Y \in \{0, 1\}^{n \times c}$ with $c$ categories, where $Y_{i,j} = 1$ indicates that the $i$-th sample has the $j$-th label, and $Y_{i,j} = 0$ indicates that the $j$-th label is not assigned to the $i$-th sample. To handle missing views, we introduce a missing-view indicator matrix $\mathcal{W} \in \{0, 1\}^{n \times m}$, where $\mathcal{W}_{i,j} = 1$ indicates that the $j$-th view of the $i$-th sample is observed, and $\mathcal{W}_{i,j} = 0$ otherwise. Similarly, we introduce a missing-label indicator matrix $\mathcal{G} \in \{0, 1\}^{n \times c}$, where $\mathcal{G}_{i,j} = 1$ means the $j$-th label of sample $i$ is observed and $\mathcal{G}_{i,j} = 0$ otherwise. Missing views and labels are filled with zeros. Our goal is to train a model for multi-label classification under the condition where both views and labels are incomplete. In this paper, $X_{i,j}$, $X_{i,:}$, and $X_{:,j}$ denote the element, the $i$-th row, and the $j$-th column of matrix $X$, respectively.

### 2.2 CONSISTENT DISCRETE REPRESENTATION LEARNING

In this section, we describe the process of learning multi-view consistent discrete representations through a shared codebook and cross-view reconstruction in three parts.

**Encoding.** Since the original dimensionalities $d_v$ of different views in multi-view data are not identical, we first use view-specific MLP encoders to map the raw data into a unified dimensional space $d_e$. Formally, $\{Z^{(v)} = E^{(v)}(X^{(v)})\}_{v=1}^m$, where $Z^{(v)} \in \mathbb{R}^{n \times d_e}$ denotes the continuous features of the $v$-th view, and $E^{(v)}$ denotes the MLP encoder of the $v$-th view.

**Quantization.** We subsequently discretize $Z^{(v)}$ through vector quantization (Van Den Oord et al., 2017), mapping each sample $Z_{i,:}^{(v)}$ from a view into a token sequence, i.e., a sequence of discrete codes. We first define a learnable shared codebook $\mathcal{V} = \{e_i\}_{i=1}^k \in \mathbb{R}^{k \times d_c}$, which contains $k$ codes, each of dimensionality $d_c$. We adopt a grouped quantization method (Baevski et al., 2019), which first splits $Z_{i,:}^{(v)}$ into $g$ segments. For clarity, taking the $i$-th sample from the $v$-th view as an example, we obtain $\tilde{Z}_{i,:}^{(v)} = [z_1, z_2, \ldots, z_g]^\top \in \mathbb{R}^{g \times (d_e/g)}$, where $z_t \in \mathbb{R}^{d_c}$ denotes the $t$-th feature segment and $d_c = d_e/g$. We assign each $z_t$ its nearest codebook embedding by nearest-neighbor lookup:

$$t^* = \arg\min_j \|\ell_2(z_t) - \ell_2(e_j)\|_2^2, \quad j = 1, \ldots, k, \tag{1}$$

Thus, we obtain the optimal quantization index $t^*$ for the $t$-th feature segment $z_t$, and denote $\hat{z}_t = e_{t^*}$, where $\ell_2(\cdot)$ represents $\ell_2$ normalization used for codebook lookup (Yu et al., 2021). Through this quantization operation, the original continuous feature $Z_{i,:}^{(v)}$ is mapped into an integer index sequence $[1^*, 2^*, \ldots, g^*] \in \{1, \ldots, k\}^g$, where each index $t^*$ corresponds to one codebook embedding. Finally, we retrieve the codebook embeddings according to these indices and concatenate them to obtain the quantized discrete representation: $\hat{Z}_{i,:}^{(v)} = [\hat{z}_1; \hat{z}_2; \ldots; \hat{z}_g] \in \mathbb{R}^{d_e}$, where $[\cdot; \cdot]$ denotes the concatenation operation. All other non-missing multi-view features $Z^{(v)}$ undergo the same quantization process to yield their discrete representations $\hat{Z}^{(v)}$.

**Reconstruction and Loss Function.** For each view, we construct a view-specific MLP decoder to reconstruct the original view $X^{(v)}$ from its discrete representation $\hat{Z}^{(v)}$, denoted as $\{D^{(v)}\}_{v=1}^m$. To better learn multi-view consistent representations, we introduce cross-view reconstruction: each view representation is decoded by different view decoders to reconstruct the original features, i.e., $\{\hat{X}^{(j,v)} = D^{(j)}(\hat{Z}^{(v)})\}_{v=1}^m, j = 1, \ldots, m$, where $\hat{X}^{(j,v)}$ denotes the reconstructed original features of view $j$ from the representation of view $v$. The reconstruction loss is defined as

$$\mathcal{L}_{rec} = \frac{1}{\sum_{i=1}^n \sum_{j=1}^m \sum_{v=1}^m \mathcal{W}_{i,j}\mathcal{W}_{i,v}} \sum_{i=1}^n \sum_{j=1}^m \sum_{v=1}^m \left\|\hat{X}_{i,:}^{(j,v)} - X_{i,:}^{(j)}\right\|_2^2 \mathcal{W}_{i,j}\mathcal{W}_{i,v} \tag{2}$$

We use an MSE-based reconstruction loss, where the missing-view indicator matrix $\mathcal{W}$ masks unavailable views. The reconstruction loss is computed only when both view $v$ and view $j$ are available, which reduces the influence of missing views on the model. Since the nearest-neighbor search in Eq 1 is non-differentiable, we follow (Van Den Oord et al., 2017) and adopt a straight-through gradient estimator: $z_t = \text{sg}[z_t - \hat{z}_t] + \hat{z}_t$, where the gradient is directly copied from the decoder input to the encoder output. The codebook learning objective is defined as

$$\underbrace{\mathcal{L}_{vq}^{(i,v)}}_{\text{sample } i, \text{ view } v} = \sum_{t=1}^g \left(\|\text{sg}[\ell_2(z_t)] - \ell_2(\hat{z}_t)\|_2^2 + \|\ell_2(z_t) - \text{sg}[\ell_2(\hat{z}_t)]\|_2^2\right), \tag{3}$$

where $\text{sg}[\cdot]$ denotes the stop-gradient operation, i.e., $\text{sg}[z] \equiv z$ and $\frac{d}{dz}\text{sg}[z] \equiv 0$. The first term forces the codebook embeddings to be close to the encoder outputs, while the second term ensures that the encoder outputs are pulled toward a codebook embedding. We compute the loss over all non-missing samples: $\mathcal{L}_{vq} = \frac{1}{\sum_{i=1}^n \sum_{v=1}^m \mathcal{W}_{i,v}} \sum_{i=1}^n \sum_{v=1}^m \mathcal{W}_{i,v} \mathcal{L}_{vq}^{(i,v)}$.

In this part, our multi-view consistent discrete representation learning consists of $m$ encoders, one quantizer, and $m$ decoders. We quantize the continuous features $\{Z^{(v)}\}_{v=1}^m$ into discrete representations $\{\hat{Z}^{(v)}\}_{v=1}^m$ using the same shared codebook. Through shared codebook quantization, the features of different views are mapped into a limited set of codebook embeddings, which not only reduces redundancy but also allows common information across views to be expressed consistently in the discrete space. Moreover, our cross-view reconstruction loss further enhances the learning of consistent multi-view representations, reducing the need for additional alignment losses.

## 2.3 CLASSIFICATION AND MULTI-VIEW DECISION FUSION

In this section, we introduce how to perform multi-label classification based on the view-consistent discrete representations $\{\hat{Z}^{(v)}\}_{v=1}^{m}$ learned in Section 2.2.

**Classification.** We first construct a multi-label classifier $F_{cls}^{(v)}(\cdot)$ for each view, which consists of a fully connected layer that maps $\hat{Z}^{(v)}$ into the label space. Formally, $\{P^{(v)} = \sigma(F_{cls}^{(v)}(\hat{Z}^{(v)})) \in \mathbb{R}^{n \times c}\}_{v=1}^{m}$, where $\sigma(\cdot)$ denotes the sigmoid activation function.

**Fusion.** Existing approaches for multi-view feature fusion and decision-level fusion mainly include average fusion, learnable weight fusion, uncertainty-aware fusion, and quality-discriminator-based fusion. Here, we propose to guide the evaluation of view prediction quality using label correlations, and then assign quantitative weights to each view prediction. Our method is more suitable for multi-view prediction fusion, as it fully exploits both multi-label supervision signals and label correlations.

Specifically, we first compute a label correlation matrix using the conditional probability matrix, following the approach in (Hang & Zhang, 2021; Chen et al., 2019). The formulation is given as

$$S_{i,j} = \frac{\sum_{r=1}^{n} Y_{r,i} Y_{r,j}}{\sum_{r=1}^{n} Y_{r,i} Y_{r,i} + \varepsilon} = \frac{Y_{:,i}^{\top} Y_{:,j}}{Y_{:,i}^{\top} Y_{:,i} + \varepsilon} \tag{4}$$

Here, $S_{i,j}$ denotes the probability of label $j$ occurring when label $i$ occurs, $\varepsilon$ denotes a small scalar. The label matrix $Y$ is taken from the training set, and the final label correlation matrix is obtained as $S \in \mathbb{R}^{c \times c}$. Next, we compute the label correlation matrix for each view prediction $\hat{P}_{r,:}^{(v)} = \mathcal{W}_{r,v} P_{r,:}^{(v)}$ in the same way:

$$S_{i,j}^{(v)} = \frac{\sum_{r=1}^{n_h} \hat{P}_{r,i}^{(v)} \hat{P}_{r,j}^{(v)}}{\sum_{r=1}^{n_h} \hat{P}_{r,i}^{(v)} \hat{P}_{r,i}^{(v)} + \varepsilon} = \frac{(\hat{P}_{:,i}^{(v)})^{\top} \hat{P}_{:,j}^{(v)}}{(\hat{P}_{:,i}^{(v)})^{\top} \hat{P}_{:,i}^{(v)} + \varepsilon} \tag{5}$$

where $n_h$ denotes the batch size at the $h$-th training step. Through this formulation, we obtain the label correlation matrices for each view, $\{S^{(v)}\}_{v=1}^{m} \in \mathbb{R}^{c \times c}$, which are computed using the predictions from the available views in the current batch. We then measure the ability of the $v$-th view to preserve label correlation structures by computing the Frobenius norm between $S^{(v)}$ and $S$, which serves as an indicator of prediction quality. Before computing the difference, we symmetrize and row-normalize both matrices to obtain $\hat{S}^{(v)}$ and $\hat{S}$. The prediction quality score and view weights are defined as

$$q^{(v)} = -\|\hat{S}^{(v)} - \hat{S}\|_F, \quad w_i^{(v)} = \frac{\exp(q^{(v)}/\tau) \cdot \mathcal{W}_{i,v}}{\sum_{u=1}^{m} \exp(q^{(u)}/\tau) \cdot \mathcal{W}_{i,u}}, \tag{6}$$

where the second term denotes the softmax normalization with a temperature parameter $\tau$. This yields the weights of all views, $\{w_i^{(v)}\}_{v=1}^{m}, i = 1, ..., n$. This method not only relies on the predictions of individual views but also explicitly leverages the global label correlation structure $S$. As a result, the weight assignment prioritizes views that align with the global label dependency patterns and reduces the influence of noisy views on the fusion results. In each batch, $S^{(v)}$ is updated according to the current predictions, so the weights adaptively reflect the relative quality of different views across training stages and batches, rather than remaining fixed.

$$P_{i,:} = \sum_{v=1}^{m} w_i^{(v)} P_{i,:}^{(v)}. \tag{7}$$

Finally, the fused prediction $P \in \mathbb{R}^{n \times c}$ is obtained by weighted fusion. We align the fused prediction $P$ with the ground-truth labels $Y$ through the binary cross-entropy loss:

$$\mathcal{L}_c = \mathcal{L}_{bce}(P, Y) = -\frac{1}{\sum_{i=1}^{n} \sum_{j=1}^{c} \mathcal{G}_{i,j}} \sum_{i=1}^{n} \sum_{j=1}^{c} \Big( Y_{i,j} \log(P_{i,j}) + (1 - Y_{i,j}) \log(1 - P_{i,j}) \Big) \mathcal{G}_{i,j}, \tag{8}$$

where the missing-label indicator matrix $\mathcal{G}$ masks the effect of missing labels on the model.

Table 1: The summary statistics of different datasets are presented, where $c$ denotes the number of classes, $c/n$ denotes the average number of positive labels per sample, $n$ denotes the number of samples, and $m$ denotes the number of views.

| Dataset | $c$ | $c/n$ | $n$ | $m$ |
|---|---|---|---|---|
| Corel5k | 260 | 3.396 | 4999 | 6 |
| Pascal07 | 20 | 1.465 | 9963 | 6 |
| Espgame | 268 | 4.686 | 20770 | 6 |
| Iaprtc12 | 291 | 5.719 | 19627 | 6 |
| Mirflickr | 38 | 4.716 | 25000 | 6 |

## 2.4 Self-Distillation-Based Prediction Enhancement

After obtaining the fused prediction $P$, we further enhance the predictive ability of the model through a self-distillation framework (Zhang et al., 2021). Specifically, we use the multi-view fused prediction $P$ as the teacher and the prediction of each individual view $P^{(v)}$ as the student, where the teacher prediction guides the learning of each student. The self-distillation loss is defined as:

$$\mathcal{L}_{\text{dis}} = \frac{1}{\sum_{i=1}^{n}\sum_{v=1}^{m}\mathcal{W}_{i,v}}\sum_{i=1}^{n}\sum_{v=1}^{m}\Big[\lambda\,\mathcal{D}_{KL}\big(\text{sg}[P_{i,:}]\,\|\,P_{i,:}^{(v)}\big) + (1-\lambda)\,\mathcal{L}_{bce}\big(P_{i,:}^{(v)},\,Y_{i,:}\big)\Big]\mathcal{W}_{i,v} \quad (9)$$

where $\lambda \in [0,1]$ denotes the imitation parameter, $\text{sg}[\cdot]$ is the stop-gradient operation defined in Section 2.2, $\mathcal{D}_{KL}$ denotes the Kullback–Leibler (KL) divergence, and $\mathcal{L}_{bce}$ is the supervision loss for each view prediction $P^{(v)}$, similar to Eq 8. Traditional distillation minimizes the KL divergence between teacher and student probabilities, assuming class probabilities sum to one. This assumption fails in multi-label learning. To address this, we adopt the multi-label logit distillation (MLD) loss (Yang et al., 2023), which follows a one-versus-all strategy by decomposing the task into binary problems and minimizing teacher–student probability differences for each, enabling effective distillation in multi-label learning.

This self-distillation framework uses the multi-view fused prediction as the teacher, which aggregates information from all views and provides a comprehensive and reliable supervisory signal. Each view-specific classifier serves as a student and learns from the teacher output, enabling it to capture the global knowledge contained in the fused prediction while preserving its own view-specific characteristics. As a result, the framework improves consistency, robustness, and generalization during both training and inference.

## 2.5 Overall Loss Function

Finally, we combine Eq 2, Eq 3, Eq 8, and Eq 9 to obtain the overall optimization objective of the model:

$$\mathcal{L} = \mathcal{L}_c + \mathcal{L}_{dis} + \alpha\mathcal{L}_{rec} + \mathcal{L}_{vq}, \quad (10)$$

where $\alpha$ is a trade-off coefficient that balances the influence of different optimization objectives. This overall objective function jointly contributes to the optimization process from the perspectives of prediction accuracy, fusion self-distillation, reconstruction quality, and representation quantization.

**Complexity Analysis.** We first define $d_{max}$ as the maximum number of neurons in the intermediate layers of the network. The computational complexities of the four loss functions $\mathcal{L}_c$, $\mathcal{L}_{dis}$, $\mathcal{L}_{rec}$ and $\mathcal{L}_{vq}$ are $O(nc)$, $O(nmc)$, $O(nm^2)$ and $O(nmg)$, respectively. The encoding–decoding stage has a time complexity of $O(nm^2 d_{max}^2)$, and the quantization process has a time complexity of $O(nmgk)$. The overall time complexity is thus $O(nm^2 d_{max}^2 + nmgk + nc + nmc + nm^2 + nmg)$. In summary, the overall computational cost of SCSD is dominated by the multi-view encoding–decoding process. The overall complexity grows linearly with the sample size $n$, and the framework exhibits good scalability in multi-view scenarios.

Table 2: The results under the setting of 50% missing views, 50% missing labels, and 70% training data are reported. The table lists mean and standard deviation (bottom-right). Ave.R denotes the average rank across metrics. Bold numbers indicate the best results, and underlined numbers indicate the second-best.

| Dataset | Metric | iMvWL IJCAI'18 | NAIM3L TPAMI'22 | DDINet TNNLS'23 | DICNet AAAI'23 | MTD NeurIPS'23 | SIP ICML'24 | RANK TPAMI'25 | DRLS CVPR'25 | SCSD — |
|---|---|---|---|---|---|---|---|---|---|---|
| Corel5k | AP | $0.283_{0.008}$ | $0.309_{0.004}$ | $0.360_{0.009}$ | $0.378_{0.004}$ | $0.413_{0.007}$ | $0.416_{0.015}$ | $0.425_{0.009}$ | $\underline{0.433}_{0.008}$ | $\mathbf{0.447}_{0.010}$ |
| | 1-HL | $0.978_{0.000}$ | $\underline{0.987}_{0.000}$ | $\underline{0.987}_{0.000}$ | $\underline{0.987}_{0.000}$ | $\mathbf{0.988}_{0.000}$ | $\mathbf{0.988}_{0.000}$ | $\mathbf{0.988}_{0.000}$ | $\mathbf{0.988}_{0.000}$ | $\mathbf{0.988}_{0.000}$ |
| | 1-RL | $0.865_{0.005}$ | $0.878_{0.002}$ | $0.865_{0.005}$ | $0.877_{0.004}$ | $0.892_{0.004}$ | $0.910_{0.003}$ | $0.913_{0.003}$ | $\underline{0.916}_{0.002}$ | $\mathbf{0.920}_{0.002}$ |
| | AUC | $0.868_{0.005}$ | $0.881_{0.002}$ | $0.868_{0.005}$ | $0.881_{0.003}$ | $0.895_{0.004}$ | $0.912_{0.003}$ | $0.915_{0.003}$ | $\underline{0.918}_{0.002}$ | $\mathbf{0.923}_{0.003}$ |
| | 1-OE | $0.311_{0.015}$ | $0.350_{0.009}$ | $0.437_{0.012}$ | $0.464_{0.012}$ | $0.491_{0.010}$ | $0.492_{0.018}$ | $0.490_{0.014}$ | $\underline{0.509}_{0.019}$ | $\mathbf{0.526}_{0.018}$ |
| | 1-Cov | $0.702_{0.008}$ | $0.725_{0.005}$ | $0.689_{0.012}$ | $0.714_{0.010}$ | $0.748_{0.009}$ | $0.786_{0.007}$ | $0.798_{0.005}$ | $\underline{0.804}_{0.006}$ | $\mathbf{0.811}_{0.006}$ |
| | Ave.R | 8.500 | 6.667 | 7.500 | 6.333 | 4.167 | 3.333 | 3.000 | 1.833 | **1.000** |
| Pascal07 | AP | $0.437_{0.018}$ | $0.488_{0.003}$ | $0.532_{0.010}$ | $0.502_{0.007}$ | $0.550_{0.004}$ | $0.550_{0.009}$ | $0.554_{0.009}$ | $\underline{0.567}_{0.008}$ | $\mathbf{0.578}_{0.009}$ |
| | 1-HL | $0.882_{0.004}$ | $0.928_{0.001}$ | $\underline{0.932}_{0.001}$ | $0.930_{0.001}$ | $\underline{0.932}_{0.001}$ | $0.931_{0.002}$ | $\underline{0.932}_{0.001}$ | $\mathbf{0.934}_{0.001}$ | $\mathbf{0.934}_{0.001}$ |
| | 1-RL | $0.736_{0.015}$ | $0.783_{0.001}$ | $0.808_{0.005}$ | $0.781_{0.007}$ | $0.830_{0.003}$ | $0.825_{0.006}$ | $0.826_{0.004}$ | $\underline{0.843}_{0.004}$ | $\mathbf{0.846}_{0.005}$ |
| | AUC | $0.767_{0.015}$ | $0.811_{0.001}$ | $0.829_{0.004}$ | $0.805_{0.006}$ | $0.849_{0.004}$ | $0.845_{0.005}$ | $0.848_{0.005}$ | $\underline{0.864}_{0.003}$ | $\mathbf{0.866}_{0.004}$ |
| | 1-OE | $0.362_{0.023}$ | $0.421_{0.006}$ | $0.448_{0.015}$ | $0.426_{0.013}$ | $0.457_{0.008}$ | $0.463_{0.012}$ | $0.465_{0.015}$ | $\underline{0.477}_{0.011}$ | $\mathbf{0.489}_{0.011}$ |
| | 1-Cov | $0.677_{0.015}$ | $0.727_{0.002}$ | $0.757_{0.005}$ | $0.728_{0.007}$ | $0.783_{0.004}$ | $0.777_{0.005}$ | $0.779_{0.005}$ | $\underline{0.798}_{0.004}$ | $\mathbf{0.801}_{0.005}$ |
| | Ave.R | 8.833 | 7.500 | 5.333 | 7.167 | 3.500 | 4.833 | 3.500 | 1.833 | **1.000** |
| Espgame | AP | $0.244_{0.005}$ | $0.286_{0.002}$ | $0.286_{0.004}$ | $0.299_{0.004}$ | $0.306_{0.003}$ | $0.310_{0.004}$ | $0.314_{0.004}$ | $\underline{0.326}_{0.005}$ | $\mathbf{0.345}_{0.004}$ |
| | 1-HL | $\underline{0.972}_{0.000}$ | $\mathbf{0.983}_{0.000}$ | $\mathbf{0.983}_{0.000}$ | $\mathbf{0.983}_{0.000}$ | $\mathbf{0.983}_{0.000}$ | $\mathbf{0.983}_{0.000}$ | $\mathbf{0.983}_{0.000}$ | $\mathbf{0.983}_{0.000}$ | $\mathbf{0.983}_{0.000}$ |
| | 1-RL | $0.808_{0.002}$ | $0.818_{0.002}$ | $0.815_{0.003}$ | $0.833_{0.003}$ | $0.837_{0.001}$ | $0.849_{0.002}$ | $0.849_{0.002}$ | $\underline{0.858}_{0.002}$ | $\mathbf{0.863}_{0.002}$ |
| | AUC | $0.813_{0.002}$ | $0.824_{0.002}$ | $0.819_{0.003}$ | $0.837_{0.002}$ | $0.842_{0.001}$ | $0.853_{0.002}$ | $0.853_{0.002}$ | $\underline{0.862}_{0.002}$ | $\mathbf{0.867}_{0.002}$ |
| | 1-OE | $0.343_{0.013}$ | $0.339_{0.003}$ | $0.427_{0.010}$ | $0.437_{0.010}$ | $0.448_{0.006}$ | $0.451_{0.012}$ | $0.460_{0.010}$ | $\underline{0.473}_{0.001}$ | $\mathbf{0.491}_{0.010}$ |
| | 1-Cov | $0.548_{0.004}$ | $0.571_{0.003}$ | $0.553_{0.005}$ | $0.598_{0.006}$ | $0.601_{0.004}$ | $0.628_{0.004}$ | $0.632_{0.005}$ | $\underline{0.652}_{0.003}$ | $\mathbf{0.657}_{0.004}$ |
| | Ave.R | 8.833 | 6.500 | 6.500 | 5.167 | 4.333 | 3.167 | 2.667 | 1.833 | **1.000** |
| Iaprtc12 | AP | $0.237_{0.003}$ | $0.261_{0.001}$ | $0.302_{0.005}$ | $0.327_{0.005}$ | $0.332_{0.002}$ | $0.331_{0.007}$ | $0.347_{0.004}$ | $\underline{0.356}_{0.006}$ | $\mathbf{0.385}_{0.005}$ |
| | 1-HL | $0.969_{0.000}$ | $\underline{0.980}_{0.000}$ | $\underline{0.980}_{0.000}$ | $\underline{0.980}_{0.000}$ | $\mathbf{0.981}_{0.000}$ | $\mathbf{0.981}_{0.000}$ | $\mathbf{0.981}_{0.000}$ | $\mathbf{0.981}_{0.000}$ | $\mathbf{0.981}_{0.000}$ |
| | 1-RL | $0.833_{0.002}$ | $0.848_{0.001}$ | $0.853_{0.002}$ | $0.872_{0.002}$ | $0.875_{0.001}$ | $0.887_{0.004}$ | $0.888_{0.002}$ | $\underline{0.896}_{0.003}$ | $\mathbf{0.903}_{0.002}$ |
| | AUC | $0.835_{0.001}$ | $0.850_{0.001}$ | $0.855_{0.003}$ | $0.873_{0.001}$ | $0.876_{0.001}$ | $0.888_{0.003}$ | $0.889_{0.002}$ | $\underline{0.898}_{0.002}$ | $\mathbf{0.905}_{0.002}$ |
| | 1-OE | $0.352_{0.008}$ | $0.390_{0.005}$ | $0.435_{0.009}$ | $0.465_{0.013}$ | $0.471_{0.006}$ | $0.466_{0.001}$ | $0.486_{0.012}$ | $\underline{0.490}_{0.012}$ | $\mathbf{0.514}_{0.008}$ |
| | 1-Cov | $0.564_{0.005}$ | $0.592_{0.004}$ | $0.594_{0.007}$ | $0.648_{0.005}$ | $0.649_{0.002}$ | $0.679_{0.008}$ | $0.686_{0.006}$ | $\underline{0.707}_{0.007}$ | $\mathbf{0.721}_{0.005}$ |
| | Ave.R | 9.000 | 7.667 | 6.833 | 6.000 | 4.000 | 3.833 | 2.667 | 1.833 | **1.000** |
| Mirflickr | AP | $0.490_{0.012}$ | $0.551_{0.002}$ | $0.588_{0.003}$ | $0.586_{0.005}$ | $0.608_{0.004}$ | $0.615_{0.004}$ | $0.606_{0.003}$ | $\underline{0.630}_{0.005}$ | $\mathbf{0.634}_{0.005}$ |
| | 1-HL | $0.839_{0.002}$ | $0.882_{0.001}$ | $0.888_{0.001}$ | $0.888_{0.001}$ | $\underline{0.891}_{0.001}$ | $\underline{0.891}_{0.001}$ | $\underline{0.891}_{0.001}$ | $\mathbf{0.895}_{0.001}$ | $\mathbf{0.895}_{0.001}$ |
| | 1-RL | $0.803_{0.008}$ | $0.844_{0.001}$ | $0.865_{0.002}$ | $0.861_{0.001}$ | $0.875_{0.001}$ | $0.878_{0.002}$ | $0.874_{0.002}$ | $\underline{0.885}_{0.002}$ | $\mathbf{0.888}_{0.002}$ |
| | AUC | $0.787_{0.012}$ | $0.837_{0.001}$ | $0.853_{0.002}$ | $0.848_{0.004}$ | $0.861_{0.002}$ | $0.864_{0.002}$ | $0.860_{0.003}$ | $\underline{0.872}_{0.003}$ | $\mathbf{0.873}_{0.002}$ |
| | 1-OE | $0.511_{0.022}$ | $0.585_{0.003}$ | $0.636_{0.008}$ | $0.642_{0.006}$ | $0.656_{0.004}$ | $\underline{0.664}_{0.006}$ | $0.654_{0.009}$ | $\mathbf{0.686}_{0.009}$ | $\mathbf{0.686}_{0.006}$ |
| | 1-Cov | $0.572_{0.013}$ | $0.631_{0.002}$ | $0.654_{0.003}$ | $0.646_{0.006}$ | $0.677_{0.002}$ | $0.676_{0.004}$ | $0.673_{0.004}$ | $\underline{0.692}_{0.003}$ | $\mathbf{0.695}_{0.004}$ |
| | Ave.R | 9.000 | 8.000 | 6.167 | 6.500 | 3.667 | 3.167 | 4.667 | 1.667 | **1.000** |

# 3 EXPERIMENTS

## 3.1 DATASETS AND METRICS

**Datasets.** We follow the experimental settings in several IMVMLC studies to comprehensively evaluate the performance of the proposed model (Liu et al., 2024b; Yan et al., 2025). We conduct experiments on five multi-view multi-label datasets, namely Corel5k (Duygulu et al., 2002), Pascal07 (Everingham et al., 2010), Espgame (Von Ahn & Dabbish, 2004), Iaprtc12 (Grubinger et al., 2006), and Mirflickr (Huiskes & Lew, 2008). More details about these datasets are provided in Table 1. We use six different types of features from these datasets as six views: DenseSift (1000), DenseHue (100), GIST (512), RGB (4096), LAB (4096), and HSV (4096), where the number in parentheses denotes the feature dimensionality.

**Evaluation Metrics.** Following previous work (Liu et al., 2023b; 2024c), we evaluate our model and all baseline methods using six commonly used metrics for multi-label classification. These include Average Precision (AP), Hamming Loss (HL), Adapted Area Under Curve (AUC), Ranking Loss (RL), OneError (OE), and Coverage (Cov). For four of these metrics, we record $1-$HL, $1-$RL, $1-$OE, and $1-$Cov in figures and tables. In this way, all six evaluation metrics follow a consistent convention: a larger value indicates better performance.

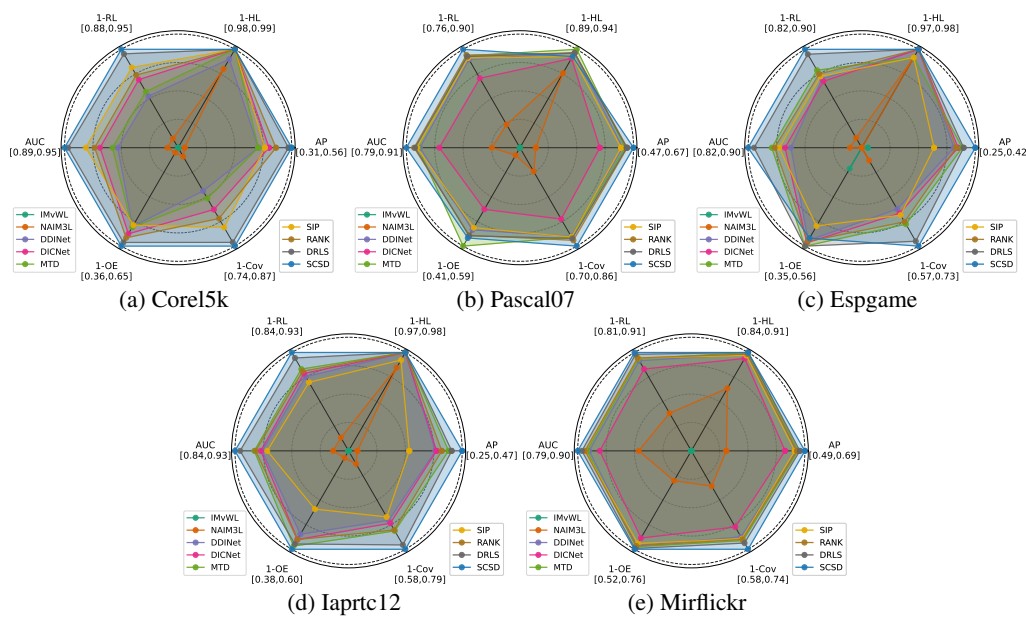

Figure 2: The radar charts are based on results with complete views, complete labels, and 70% training data, covering nine methods, five datasets, and six metrics. In each chart, the center denotes the worst result and the vertex denotes the best.

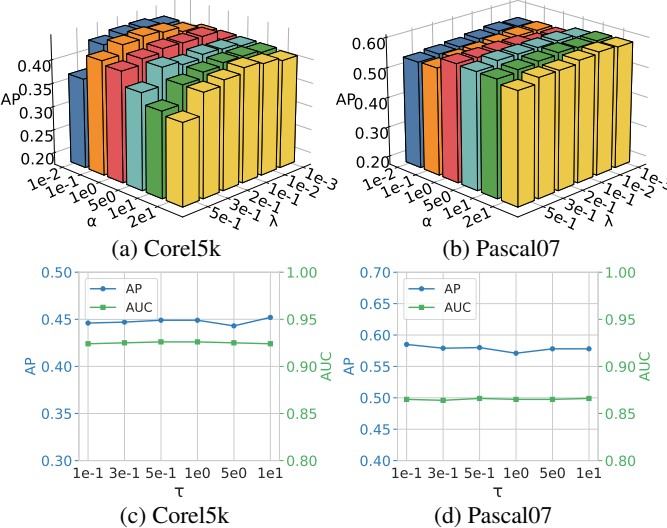

Figure 3: The parameter sensitivity analysis of the SCSD model is conducted under the setting of 50% missing views, 50% missing labels, and 70% training data.

## 3.2 COMPARISON METHODS

To more comprehensively evaluate the effectiveness of the proposed method, we select eight incomplete multi-view multi-label learning methods specifically designed for the dual-missing problem as baselines in the comparative experiments. This enables a more comprehensive evaluation of the model's ability to handle dual-missing scenarios. The specific methods include iMvWL (Tan et al., 2018), NAIM3L (Li & Chen, 2021), DDINet (Wen et al., 2023), DICNet (Liu et al., 2023b), MTD (Liu et al., 2024b), SIP (Liu et al., 2024c), RANK (Liu et al., 2025), and DRLS (Yan et al., 2025), whose related descriptions are already provided in the Introduction 1 and Related Work A.1 sections.

## 3.3 IMPLEMENTATION DETAILS

To simulate the random missingness of multi-view and multi-label data in real-world scenarios, we follow previous studies to generate missing data (Tan et al., 2018; Liu et al., 2024c). Specifically, for

multi-view data, we randomly discard 50% of the views while ensuring that each sample retains at least one available view. For multi-label data, we randomly discard 50% of the positive and negative labels, and we use zeros to fill in the missing views and labels. The dataset is divided into 70% for training and 30% for validation and testing. The proposed SCSD model is implemented in PyTorch and the experiments are conducted on an Ubuntu operating system with an RTX 4090 GPU and an i9-13900K CPU. The learning rate is set to 0.001, the optimizer is AdamW with a weight decay of 0.001, and the batch size is 128. The codebook is initialized with k-means, the codebook size $k$ is set to 2048, and the codebook embedding dimension $d_c$ is set to 4.

### 3.4 EXPERIMENTAL RESULTS

Table 2 compares eight state-of-the-art methods on five public multi-view multi-label datasets, with both view and label missing rates set to 50%. It can be observed that the proposed SCSD model outperforms all baseline methods, especially on the AP metric of the Espgame and Iaprtc12 datasets, where SCSD achieves improvements of 5.83% and 8.15% over the second-best method, DRLS. The Espgame and Iaprtc12 datasets have more complex label spaces, which introduce a higher level of learning difficulty. Achieving significant improvements under these more challenging label structures indicates that SCSD has a stronger capability to model complex multi-label relationships and learn cross-view consistency. Compared with DICNet, which learns multi-view consistent features through contrastive loss, and SIP, which suppresses non-shared information based on the information bottleneck principle to obtain consistent representations, the proposed SCSD achieves average improvements of 14.94% and 8.65% in AP across the five datasets. These results clearly demonstrate the advantage of SCSD in multi-view consistent representation learning. This comparative experiment thoroughly validates the effectiveness of SCSD for multi-label classification under the dual-missing scenario.

In addition, we also conduct comparative experiments under the setting of complete views and complete labels, as shown in Figure 2. It can be observed that SCSD achieves the best performance on most metrics across five datasets, which strongly demonstrates the generality of SCSD. Under the condition where both views and labels are complete, SCSD still achieves the best or near-best performance on most metrics. This suggests that the consistency representation learning mechanism built on the multi-view shared codebook has strong inherent representational capacity, and its effectiveness is not limited to cases with missing information.

### 3.5 PARAMETER ANALYSIS

Our model contains three hyperparameters: $\alpha$ in $\mathcal{L}_{rec}$, $\lambda$ in $\mathcal{L}_{dis}$, and the softmax temperature parameter $\tau$ in decision fusion. Figure 3 presents the parameter sensitivity results of the SCSD model. Figures 3a and 3b show the AP metric of SCSD on Corel5k and Pascal07 under different combinations of $\alpha$ and $\lambda$. We observe that on the Corel5k dataset, SCSD exhibits performance fluctuations when $\alpha = 1e-2$ or $\alpha = 2e1$, which are extreme values, while on Pascal07 the performance of SCSD remains relatively stable. On Corel5k, the best results are obtained when $\alpha$ takes values in the range $[1e-2, 1e0]$ and $\lambda$ takes values in the range $[1e-2, 2e-1]$, whereas on Pascal07, better performance is achieved when $\alpha$ takes values in the range $[5e0, 2e1]$ and $\lambda$ takes values in the range $[1e-2, 2e-1]$. Figures 3c and 3d present the influence of $\tau$ on the model, where the left $y$-axis indicates AP and the right $y$-axis indicates AUC. The proposed method is not sensitive to variations of the temperature parameter $\tau$. On Corel5k, $\tau$ takes values in the range $[5e-1, 5e0]$ to achieve the best results, while on Pascal07, $\tau$ takes values in the range $[1e-1, 5e-1]$ for the best performance.

### 3.6 ABLATION STUDY

Table 3 presents the ablation study of SCSD, where the gray background in the middle highlights the full version of SCSD. The upper part removes different loss functions. Among them, $\mathcal{L}_{dis\_KL}$ denotes the first term in $\mathcal{L}_{dis}$, which encourages the student to imitate the output of the fused teacher. We observe that removing any loss function leads to a performance drop of SCSD. The lower part of the table removes certain structural designs. In the fifth row, "w/o VQ" indicates that vector quantization is not used, and the continuous features $\{Z^{(v)}\}_{v=1}^{m}$ output by the encoder are directly employed. A clear performance drop is observed, since our multi-view shared codebook design

Table 3: The ablation results on two datasets under the setting of 50% missing views, 50% missing labels, and 70% training data are reported. Here, "w/o" denotes "without". The bold numbers indicate the best results, while the underlined numbers indicate the second-best results.

| Method | Corel5k | | | Pascal07 | | |
|---|---|---|---|---|---|---|
| | AP | 1-RL | AUC | AP | 1-RL | AUC |
| SCSD w/o $\mathcal{L}_{dis}$ | 0.376 | 0.882 | 0.884 | 0.560 | 0.834 | 0.855 |
| SCSD w/o $\mathcal{L}_{dis\_KL}$ | 0.411 | 0.906 | 0.909 | 0.572 | 0.843 | 0.864 |
| SCSD w/o $\mathcal{L}_{rec}$ | 0.439 | 0.916 | 0.919 | 0.560 | 0.839 | 0.860 |
| **SCSD** | **0.447** | **0.920** | **0.923** | **0.578** | **0.846** | **0.866** |
| SCSD w/o VQ | 0.430 | 0.914 | 0.916 | 0.565 | 0.841 | 0.860 |
| SCSD w/o cross_view_rec | 0.442 | 0.918 | 0.921 | 0.553 | 0.837 | 0.859 |
| SCSD w/o S_fusion | 0.445 | 0.919 | 0.922 | 0.570 | 0.844 | 0.864 |

better supports consistent representation learning. In the sixth row, "w/o cross_view_rec" denotes removing cross-view reconstruction and training with standard single-view reconstruction, which also results in performance degradation to some extent. The last row, "w/o S_fusion," denotes removing our weighted fusion strategy and replacing it with a simple masked average fusion strategy: $P_{i,:} = (\sum_{v=1}^{m} P_{i,:}^{(v)} \mathcal{W}_{i,v}) / \sum_{v=1}^{m} \mathcal{W}_{i,v}$, where we observe a performance decline, especially on the Pascal07 dataset. This is because Pascal07 has 20 labels, which provide a more reliable label correlation matrix $S$, enabling our fusion strategy to better identify the quality of predictions from different views. Overall, we find that the contributions of the multi-view shared codebook and self-distillation are the most significant for the performance of SCSD.

# 4 CONCLUSION

In this paper, we propose a novel method for incomplete multi-view multi-label classification. First, we use a multi-view shared codebook to learn consistent discrete representations across views, and we further enhance the consistency of different view representations through a cross-view reconstruction mechanism. Then, we allocate different weights by evaluating the ability of each view prediction to preserve label correlation structures, and we perform weighted fusion to obtain the fused prediction. Finally, we use the fused prediction as the teacher to guide the learning of each view prediction, and we feed the knowledge of all views back into each view-specific branch through the self-distillation loss, thereby improving the generalization ability of the model. Extensive experiments demonstrate that the SCSD method effectively addresses the problem of multi-view multi-label classification under dual-missing conditions.

**Limitations.** Although our method achieves strong performance on the incomplete multi-view multi-label learning task, it still has several limitations that may affect its applicability in broader scenarios. First, introducing a multi-view shared codebook brings additional memory and computational overhead. The memory overhead mainly comes from storing and updating the codebook embeddings, while the computational cost is largely due to computing the distance matrix between input features and the codebook embeddings during quantization. In addition, the quantization modules in SCSD assume that representations from different views can be aligned in a shared latent space. When the view-missing rate becomes very high, the amount of cross-view information available for alignment is greatly reduced, which can weaken the generalization ability of the shared codebook mechanism.

REPRODUCIBILITY STATEMENT

All experiments in this paper are conducted on five publicly available multi-view multi-label datasets, ensuring that no private or proprietary data are used. The pseudocode of the training procedure is provided in Appendix A.2. The source code of SCSD is available on GitHub.

ACKNOWLEDGMENTS

This work was supported by the National Natural Science Foundation of China (Grant No. 62476166 and No. 62576206).

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

# A  APPENDIX

## A.1  RELATED WORK

**Multi-View Multi-Label Learning**. Under complete views and labels, several representative methods have been proposed. SIMM (Wu et al., 2019) jointly optimizes a confusion-adversarial loss and a multi-label loss to exploit shared information, while imposing orthogonal constraints on the shared subspace to preserve discriminative features. To explicitly address both view consistency and diversity, CDMM (Zhao et al., 2021) models view consistency with independent classifiers, incorporates the Hilbert–Schmidt independence criterion to capture diversity, and introduces label correlations and view contribution factors to enhance performance. From a graph-based perspective, D-VSM (Lyu et al., 2022) encodes view features with deep GCNs and integrates cross-view relations within a unified graph. Moreover, focusing on label dependency modeling, ELSMML (Liu et al., 2023a) constructs a label correlation matrix using high-order strategies, combines dimensionality reduction to extract latent semantic features, introduces manifold regularization to preserve structural information, and trains classifiers with an accelerated optimization algorithm.

**Incomplete Multi-View Multi-Label Learning**. To handle missing views and labels, several methods have been developed for incomplete multi-view multi-label learning. iMvWL (Tan et al., 2018) learns cross-view relationships and weak label information simultaneously in the shared subspace, while capturing local label correlations and learning the corresponding predictors. To tackle label insufficiency and view misalignment under incomplete settings, NAIM3L (Li & Chen, 2021) alleviates label insufficiency through consistency constraints and label structure modeling, and jointly models both global and local structures in a common label space. From a network architecture perspective, DDINet (Wen et al., 2023) consists of feature extraction, weighted fusion, classification, and decoding modules, effectively integrating available data and labels under dual-missing scenarios. By introducing a masked mechanism, MTD (Liu et al., 2024b) proposes a masked dual-channel disentanglement framework that separates representations into shared and private channels, and enhances feature learning with contrastive loss and graph regularization. Moreover, focusing on representation disentanglement, DRLS (Yan et al., 2025) extracts shared features via cross-view reconstruction, learns view-specific features with mutual information constraints, and leverages label correlations to guide semantic embeddings for preserving topological structures.

---

**Algorithm 1:** The training process of SCSD

---

**Input:** Incomplete multi-view data $\{X^{(v)}\}_{v=1}^m$, incomplete label matrix $Y$, missing-view indicator matrix $\mathcal{W}$, missing-label indicator matrix $\mathcal{G}$, hyperparameters $\alpha$, $\lambda$, and $\tau$, and training epochs $H$.

**Output:** Prediction P.

1  Initialize the model parameters. Use Eq 4 to compute the label correlation matrix $S$. Set $codebook\_initialized$ = False.

2  **for** $h = 1$ **to** $H$ **do**

3      Extract multi-view continuous features $\{Z^{(v)} = E^{(v)}(X^{(v)})\}_{v=1}^m$ through the encoders.

4      Split the non-missing features $\{Z^{(v)}\}_{v=1}^m$ into feature segments
    $\{\tilde{Z}_{i,:}^{(v)} = [z_1, z_2, \ldots, z_g]^\top \in \mathbb{R}^{g \times (d_e/g)} \mid i = 1, \ldots, n,\ v = 1, \ldots, m,\ \mathcal{W}_{i,v} \neq 0\}$.

5      **if** *not codebook\_initialized* **then**

6          Use all available view features $\{\tilde{Z}_{i,:}^{(v)}\}$ within the current batch to perform k-means clustering for initializing the codebook embeddings.

7          $codebook\_initialized$ = True

8      Use Eq 1 to find the nearest codebook embedding $e_{t*}$ for each $z_t$, and concatenate them to obtain the discrete features $\{\hat{Z}^{(v)}\}_{v=1}^m$.

9      Obtain the cross-view reconstruction results through the decoders:
    $\{\hat{X}^{(j,v)} = D^{(j)}(\hat{Z}^{(v)})\}_{v=1}^m, j = 1, \ldots, m$.

10      Obtain the predictions of each view through the classifiers: $\{P^{(v)} = \sigma(F_{cls}^{(v)}(\hat{Z}^{(v)}))\}_{v=1}^m$.

11      Compute the weights according to Eq 5, 6, 7 and obtain the fused multi-view prediction $P$.

12      Compute the overall loss $\mathcal{L}$ according to Eq 10 and update the parameters.

---

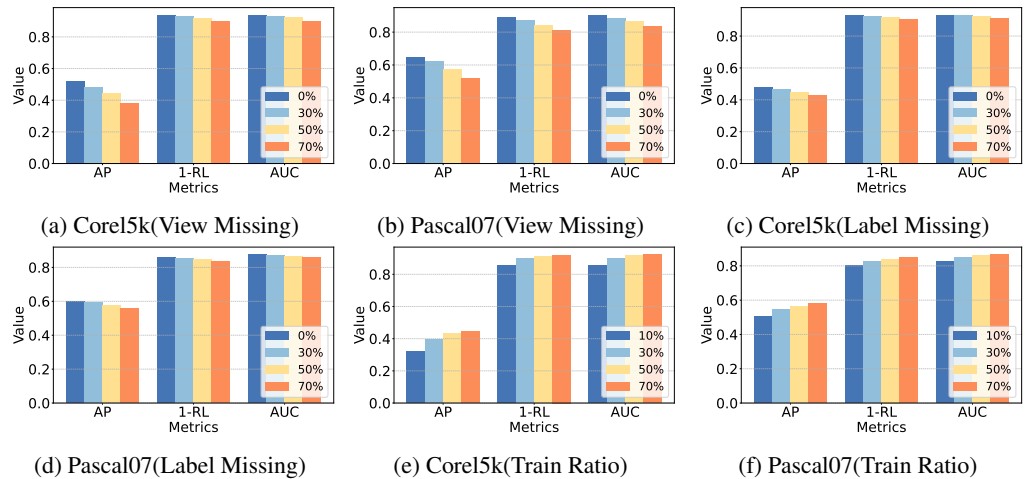

Figure 4: The experimental results of the SCSD model under different view-missing rates, different label-missing rates, and different training set proportions are reported. The figure presents two datasets and three evaluation metrics.

## A.2 ALGORITHM

The training procedure of the SCSD model is provided in Algorithm 1.

## A.3 ADDITIONAL EXPERIMENTAL RESULTS

**Missing and Training Sample Rates Analysis.** Figures 4a and 4b show the results of the SCSD model under different view-missing rates when the label-missing rate is fixed at 50%. Figures 4c and 4d present the results under different label-missing rates when the view-missing rate is fixed at 50%. As the view-missing rate or the label-missing rate gradually increases, the model performance also decreases. However, our model is able to maintain relatively stable performance even when the missing rate reaches 70%. Moreover, we observe that increasing the view-missing rate has a greater impact on our model than increasing the label-missing rate. This is because our model relies on the learned multi-view consistent representations, and the quality of the learned representations decreases when the view-missing rate increases. Figures 4e and 4f show the results of SCSD under 50% missing views and 50% missing labels with different proportions of the training set. As the proportion of the training set increases, the model performance also improves. Furthermore, our model achieves a satisfactory result even under the extreme case of only 10% training data.

**Additional Parameter Analysis.** In Figures 5a and 5b, we also retrain and evaluate the model with different scales of the codebook size $k$ and the codebook embedding dimension $d_c$ to systematically analyze the impact of the multi-view shared codebook on representation capacity and final performance. The experimental results show that appropriately increasing the codebook size improves the model performance within a certain range, but overly large values lead to higher computational cost with diminishing returns. At the same time, a smaller embedding dimension stabilizes the quantization process and improves codebook utilization, thereby leading to better model performance.

**Codebook Utilization Analysis.** Figure 6 shows the changes in codebook utilization of the SCSD model on the validation set during the training process. In this codebook utilization experiment, we compute the utilization rate by counting the number of codebook embeddings that are actually selected during the forward pass and dividing it by the total codebook size; codebook embeddings that are not assigned to any input features are regarded as inactive. This approach intuitively reflects how well the model covers the codebook prototypes during the quantization stage and indicates the efficiency of the model. We only present 10 epochs, because afterward all datasets maintain 100% codebook utilization until the end of training. From the figure, we observe that SCSD reaches 100% codebook utilization within only a few epochs on all datasets and keeps it stable throughout the subsequent training. This indicates that SCSD is able to fully activate all embedding units in the shared codebook, thereby avoiding the codebook collapse problem (i.e., only a very small number of codebook embeddings are frequently used while most vectors remain idle and inactive,

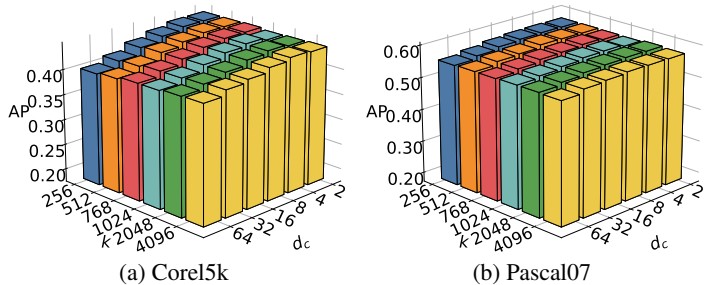

(a) Corel5k  (b) Pascal07

Figure 5: An additional parameter sensitivity analysis of the SCSD model is conducted under the setting of 50% missing views, 50% missing labels, and 70% training data.

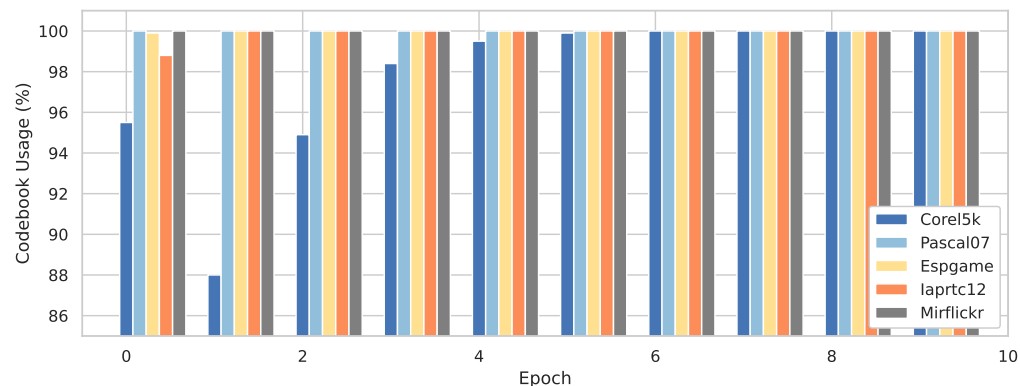

Figure 6: The codebook utilization of the SCSD method is reported under the training setting of 50% missing views, 50% missing labels, and 70% training data, covering all five datasets.

leading to insufficient representation capacity and low information utilization). In other words, the shared codebook design of SCSD not only preserves the rich representational capacity of multi-view data but also effectively suppresses redundant features through a limited number of codebook embeddings, thereby enhancing the generalization ability of the learned representations.

## A.4 LARGE LANGUAGE MODEL USAGE STATEMENT

In this paper, we use a large language model to polish the introduction section.

