# OpenReview forum: "Incomplete Multi-View Multi-Label Classification via Shared Codebook and Fused-Teacher Self-Distillation"
_ICLR.cc/2026/Conference — ICLR 2026 Poster_

### Official Review · Reviewer_gK1A · 2025-10-29

**Soundness:** 2
**Presentation:** 3
**Contribution:** 3
**Rating:** 4
**Confidence:** 3

**Summary:**

This paper proposes a novel framework, SCSD, designed to address incomplete multi-view data scenarios that involve both missing views and missing labels. The approach incorporates three core components: (1) a shared codebook that facilitates the learning of discrete and consistent representations by aligning heterogeneous views within a compact latent space; (2) an adaptive view-weight estimation mechanism that leverages label correlations to enhance the reliability of fused predictions; and (3) a fusion-guided self-distillation strategy that transfers holistic knowledge from the fused predictions back to individual view branches, thereby improving generalization and strengthening inter-view collaboration.

**Strengths:**

1.The SCSD framework enhances the learning process from three complementary aspects—representation learning, decision-level fusion, and training strategy—resulting in a well-organized and conceptually coherent design.
2.The model exhibits strong scalability, and its discrete representation learning module can be easily adapted or extended to other tasks and architectures.
3.The ablation studies clearly demonstrate that the fusion-teacher self-distillation mechanism substantially boosts classification accuracy, validating its effectiveness.

**Weaknesses:**

1.While the framework integrates several functional modules that could affect computational efficiency, the paper does not provide a time complexity or efficiency analysis. Including such an evaluation would offer readers a clearer picture of the model’s practical scalability.
2.The codebook utilization analysis in the appendix lacks details on how the utilization rate is quantified, which reduces reproducibility.
3.In the experimental section, the authors primarily report superior results over baselines but do not offer deeper performance interpretation or insight. A more thorough discussion would enhance the paper’s contribution.
4.Since SCSD jointly addresses both missing views and missing labels, it would be helpful to discuss whether any trade-off exists between these two factors—specifically, whether the model exhibits higher sensitivity to one form of incompleteness than the other.

**Questions:**

See the weaknesses above.

---

> ### Author Response · Authors · 2025-11-21
> **Response to Reviewer gK1A Part I**
>
> We thank the reviewer for the insightful comments and concrete suggestions, which allow us to re-examine the study and significantly improve the quality of the paper.
>
> **Time complexity of the SCSD model.**
>
> We appreciate the reviewer’s valuable suggestion. Regarding the computational efficiency of the model, we add a corresponding time complexity analysis in the revised manuscript to help readers more comprehensively understand the scalability and computational cost of SCSD in practical applications. Specifically, we analyze the computational cost of the main modules and loss functions of the model, and the related content is presented as follows:
>
> Complexity Analysis. We first define $d_{max}$ as the maximum number of neurons in the intermediate layers of the network. The computational complexities of the four loss functions $\mathcal{L}\_c$, $\mathcal{L}\_{dis}$, $\mathcal{L}\_{rec}$ and $\mathcal{L}\_{vq}$ are $O(nc^2)$, $O(nmc^2)$, $O(nm^2)$ and $O(nmg)$, respectively. The time complexities of the encoder and decoder are $O(nm^2d_{max}^2)$, and the quantization process has a time complexity of $O(nmgk)$. The overall time complexity is thus $O(nm^2d^2_{max}+nmgk+nc^2+nmc^2+nm^2+nmg)$. In summary, the overall computational cost of SCSD is dominated by the multi-view encoding–decoding process, while the quantization and distillation modules introduce only minor additional overhead. The overall complexity grows linearly with the sample size $n$, and the framework exhibits good scalability in multi-view scenarios.
>
> The above content is added to Section 2.5 of the main text.
>
> **The computation of the codebook utilization experiment in the appendix.**
>
> In this codebook utilization experiment, we compute the utilization rate by counting the number of codebook vectors that are actually selected during forward inference and dividing it by the total codebook size; codebook embeddings that are not assigned to any input features are regarded as inactive. This approach intuitively reflects how well the model covers the codebook prototypes during the quantization stage and indicates the efficiency of the model.
>
> The related computation method is further supplemented in the appendix of the revised manuscript.

---

> ### Author Response · Authors · 2025-11-21
> **Response to Reviewer gK1A Part II**
>
> **The experimental analysis is not sufficiently in-depth.**
>
> We appreciate the reviewer’s suggestion. The analysis of the comparative experiments in the original version is indeed somewhat brief, and in the revised manuscript we provide a more thorough discussion of the experimental results from multiple perspectives. Specifically, we supplement an analysis of the characteristics of different datasets and explain why SCSD achieves larger performance gains on datasets with more complex label spaces (such as Espgame and Iaprtc12). At the same time, we further discuss how SCSD still achieves the best or near-best performance under the setting of complete views and complete labels, indicating that its consistency-aware representation learning mechanism demonstrates robust modeling capability in both complete and incomplete scenarios. In addition, we add a systematic analysis of the shared codebook size and the embedding dimension, providing a more comprehensive explanation of their effects on the final performance from the perspectives of model capacity, computational cost, and representation ability. Through these extended discussions, we aim to offer readers more substantial performance interpretation and deeper experimental insights.
>
> The above content is added and improved in the corresponding sections of the revised manuscript, including the analysis of experimental results (Section 3.4) and the parameter analysis. Newly added content is highlighted in blue in the revised version for the reviewer’s convenience.
>
> **Whether the model makes a trade-off between missing views and missing labels when both types of incompleteness occur, and whether it is more sensitive to one type of missingness.**
>
> To examine the potential trade-off between view missingness and label missingness, we conduct two controlled experiments in the appendix: one gradually increases the label-missing rate while keeping the view-missing rate fixed, and the other gradually increases the view-missing rate while keeping the label missing-rate fixed. The experimental results are shown in the bar charts in Figures 4a–4d, and below we provide a preview table that includes only the AP metric, where the numbers in parentheses indicate the drop relative to the previous entry.
>
> label-missing rates = 50%
>
> | dataset  | view-missing rates |                          |                          |                          |
> | -------- | ------------------ | ------------------------ | ------------------------ | ------------------------ |
> |          | 0%                 | 30%                      | 50%                      | 70%                      |
> | Corel5k  | 0.520              | 0.486($\downarrow$0.034) | 0.447($\downarrow$0.039) | 0.384($\downarrow$0.063) |
> | Pascal07 | 0.650              | 0.622($\downarrow$0.028) | 0.578($\downarrow$0.044) | 0.523($\downarrow$0.055) |
>
>  view-missing rates = 50%
>
> | dataset  | label-missing rates |                          |                          |                          |
> | -------- | ------------------- | ------------------------ | ------------------------ | ------------------------ |
> |          | 0%                  | 30%                      | 50%                      | 70%                      |
> | Corel5k  | 0.480               | 0.467($\downarrow$0.013) | 0.447($\downarrow$0.020) | 0.430($\downarrow$0.017) |
> | Pascal07 | 0.598               | 0.592($\downarrow$0.006) | 0.578($\downarrow$0.014) | 0.556($\downarrow$0.022) |
>
> The results show that increasing the view missing rate has a greater impact on the performance of the SCSD model than increasing the label missing rate.
>
> This is because the core design of SCSD relies on learning robust multi-view consistency representations to support classification. As the view-missing rate increases, the cross-view information available for alignment and consistency construction drops significantly, which reduces the effectiveness of the shared codebook and the distillation mechanism and leads to a decline in representation quality. In contrast, although missing labels limit the amount of supervision, the model can still form relatively stable representations through the shared codebook and cross-view reconstruction, so the performance degradation caused by label missingness is generally more moderate.
>
> The related discussion is included in the appendix of the revised manuscript, where the experiments explicitly demonstrate the model’s different sensitivities to different types of incompleteness and explain the underlying mechanisms.

---

### Official Review · Reviewer_Z7Gz · 2025-10-30

**Soundness:** 3
**Presentation:** 3
**Contribution:** 3
**Rating:** 8
**Confidence:** 5

**Summary:**

This paper addresses the challenge that existing incomplete multi-view multi-label classification methods fail to effectively capture stable and discriminative shared semantic representations. To overcome this limitation, the authors propose a novel framework named Incomplete Multi-View Multi-Label Classification via Shared Codebook and Fused-Teacher Self-Distillation (SCSD). The SCSD model learns discrete and consistent representations across multiple views through a shared codebook and a cross-view reconstruction mechanism. Moreover, a view-weight estimation strategy is introduced to adaptively assess the relative importance of different views, thereby improving the quality of the fused prediction. In addition, a fused-teacher self-distillation framework is designed to enhance the generalization capability of the model. Extensive experiments conducted on multiple benchmark datasets demonstrate the superior performance and effectiveness of SCSD compared with existing state-of-the-art methods.

**Strengths:**

1.	The proposed approach effectively integrates a shared codebook and cross-view reconstruction to learn consistent discrete representations across multiple views. The method is conceptually well-founded and supported by solid experimental evidence.
2.	The weight estimation strategy and a teacher self-distillation framework are introduced to improve the quality of view fusion and the generalization of the proposed model.
3.	Comprehensive experiments on five publicly available benchmark datasets show that SCSD achieves superior performance across various evaluation metrics, confirming its robustness and effectiveness.

**Weaknesses:**

1.	In Figures 3(a) and 3(b) (page 8), the axis labels are too small, which negatively affects readability. Enlarging the font size would improve the presentation quality.
2.	In the overall optimization objective (Eq. 10), four different loss terms are included, but a weighting coefficient is applied only to the third one. The reasoning behind this particular design choice should be further clarified.
3.	The process for determining the size of the shared codebook is not clearly explained. It would be valuable to discuss whether increasing the codebook size consistently improves performance or whether the gains plateau beyond a certain point.
4.	The manuscript would benefit from a more detailed discussion of the limitations of the proposed approach, which would make the evaluation of SCSD more balanced and comprehensive.

**Questions:**

Refer to the weaknesses above.

---

> ### Author Response · Authors · 2025-11-21
> **Response to Reviewer Z7Gz**
>
> We thank the reviewer for the careful reading of our manuscript and the constructive comments. We revise the paper accordingly.
>
> **Regarding the issue of small font size in the figures.**
>
> We appreciate the reviewer’s suggestion. In the revised manuscript, we adjust the font size of the axes in the figures to ensure better readability both on screen and in printed form. We also unify the overall layout and visual proportions of the figures to improve their presentation quality.
>
> These revisions are reflected in Figure 3 of the updated manuscript.
>
> **The setting of the weighting coefficients in Eq.10.**
>
> This is because our preliminary experiments show that the SCSD model is not very sensitive to the balance between the distillation loss $\mathcal{L}\_{dis}$ and the codebook loss $\mathcal{L}\_{vq}$. For this reason, we simply fix their coefficients to 1 to avoid unnecessary hyperparameter tuning. In addition, $\mathcal{L}\_{dis}$ already includes the imitation weight $\lambda$, which naturally balances the student’s alignment with the teacher distribution and its learning from the ground-truth labels. In prior related work, the classification loss $\mathcal{L}\_{c}$ is typically regarded as the main factor that guides the optimization. Following this common practice, we introduce the coefficient $\alpha$ only for the reconstruction loss $\mathcal{L}\_{rec}$ so that its effect can be controlled more flexibly.
>
> **Regarding the choice of codebook size and the tuning process—does using a larger codebook necessarily lead to better performance?**
>
> In the revised manuscript, we add an experiment with different codebook sizes (Figures 3c and 3d) to systematically evaluate the impact of the shared codebook scale on model performance. The experimental results on the Corel5k dataset (see the AP results in the table below) show that the model performance does not continuously improve as the codebook size increases. When the codebook is relatively small (e.g., $k = 256$ or $512$), moderately enlarging the codebook enhances the representation capacity and thus brings certain performance gains. However, when the codebook size continues to grow into a larger range (e.g., $k \ge 1024$), the improvement saturates, and in some configurations even slightly decreases. This indicates that an excessively large codebook may introduce redundant prototypes, increase quantization instability, and weaken the model’s ability to aggregate cross-view consistent semantics.
>
> Overall, $k = 2048$ maintains relatively stable and competitive performance across multiple embedding dimensions. It provides sufficient discrete expressiveness while avoiding the redundancy and overfitting risks brought by an excessively large codebook. Therefore, we select 2048 as the shared codebook size in the final model, which represents a balanced and reasonable choice considering both performance and model complexity.
>
> | dataset  | k     |       |       |       |       |       |
> | -------- | ----- | ----- | ----- | ----- | ----- | ----- |
> |          | 256   | 512   | 768   | 1024  | 2048  | 4096  |
> | Corel5k  | 0.445 | 0.445 | 0.446 | 0.448 | 0.447 | 0.447 |
> | Pascal07 | 0.576 | 0.576 | 0.578 | 0.578 | 0.578 | 0.575 |
>
> **Lack of a systematic analysis of the limitations of the SCSD model.**
>
> In the conclusion section of the main text, we add a discussion of the limitations of the SCSD model, mainly emphasizing its constraints in terms of computational cost, sensitivity to view missingness, and the assumption of latent-space alignment. The specific content is presented as follows:
>
> Limitations. Although our method achieves strong performance on the incomplete multi-view multi-label learning task, it still has several limitations that may affect its applicability in broader scenarios. First, introducing a multi-view shared codebook brings additional spatial and computational overhead. The spatial cost mainly comes from storing and updating the codebook embedding vectors, while the computational cost is largely due to computing the distance matrix between input features and the codebook embeddings during quantization. In addition, the quantization modules in SCSD assume that representations from different views can be aligned in a shared latent space. When the view-missing rate becomes very high, the amount of cross-view information available for alignment is greatly reduced, which can weaken the generalization ability of the shared codebook mechanism.

---

### Official Review · Reviewer_3pou · 2025-10-30

**Soundness:** 3
**Presentation:** 3
**Contribution:** 3
**Rating:** 6
**Confidence:** 4

**Summary:**

This paper proposes the SCSD framework, which aims to learn consistent multi-view representations under missing-view scenarios. The method enforces representation consistency through a shared cross-view codebook, strengthens semantic alignment via cross-view reconstruction, and adaptively refines view-specific weights based on label correlations. By integrating these components into a unified architecture, the proposed approach achieves robust and reliable performance across various benchmark datasets, demonstrating both novelty and promising experimental potential.

**Strengths:**

1.	The study addresses the dual-missing problem in multi-view multi-label learning, a challenging and practically significant research direction that has received limited prior exploration.
2.	The proposed shared codebook mechanism effectively mitigates instability in semantic consistency learning under missing-view conditions, outperforming existing methods theoretically and empirically.
3.	The experimental design is rigorous and systematic, and the ablation studies clearly validate the contribution of each component within the SCSD framework.

**Weaknesses:**

1.	Although Table 2 presents extensive quantitative results, the corresponding analysis in Section 3.4 is relatively concise. A more detailed discussion from multiple perspectives (e.g., robustness, scalability, or dataset characteristics) would improve the interpretability of the findings.
2.	The overall loss function in the method section comprises several components. Adding a concise explanatory paragraph summarizing their roles would help readers grasp the optimization objective more intuitively.
3.	While the use of a shared codebook to learn discrete representations is effective, the paper offers insufficient intuitive explanation or conceptual justification for why discretization enhances performance. Providing such insight would strengthen the theoretical grounding of the approach.
4.	Given the potential heterogeneity of feature distributions across different views, it would be valuable to discuss whether the shared codebook design might unduly restrict view-specific representations, potentially leading to semantic compression or information loss.

**Questions:**

Please refer to the Weaknesses.

---

> ### Author Response · Authors · 2025-11-21
> **Response to Reviewer 3pou**
>
> We sincerely thank the reviewer for the constructive suggestions and the careful evaluation of our manuscript. These comments help us further refine and improve the study.
>
> **Analysis of the results in Table 2.**
>
> We appreciate the reviewer’s suggestion. In the revised manuscript, we enrich the experimental analysis in Section 3.4 and provide a more in-depth discussion of the results in Table 2 from multiple perspectives. We further supplement the analysis from the viewpoints of dataset characteristics and model generalization ability: on the one hand, the Espgame and Iaprtc12 datasets have more complex label spaces, and achieving significant improvements in such more challenging scenarios shows that SCSD has stronger capability in modeling multi-label relationships and cross-view consistency; on the other hand, under the setting where both views and labels are complete, SCSD still maintains leading performance on most metrics, indicating that its consistency-aware representation learning mechanism demonstrates robust modeling ability in both complete and incomplete scenarios.
>
> The corresponding additions and improvements have been incorporated into Section 3.4 of the revised manuscript.
>
> **Explanation of the sub-objectives in the overall objective function.**
>
> Our overall objective function is $\mathcal{L} = \mathcal{L}\_c + \mathcal{L}\_{dis} + \alpha \mathcal{L}\_{rec} + \mathcal{L}\_{vq}$, where $\mathcal{L}\_c$ denotes the classification loss, $\mathcal{L}\_{dis}$ denotes the distillation loss, $\mathcal{L}\_{rec}$ denotes the cross-view reconstruction loss, and $\mathcal{L}\_{vq}$ is used to learn the codebook embeddings. These loss terms jointly contribute to the overall optimization objective from the perspectives of prediction accuracy, fusion self-distillation, reconstruction quality, and representation quantization.
>
> We add a brief overview in the revised manuscript to help readers more intuitively understand the roles of these loss components.
>
> **Why the shared codebook mechanism brings performance improvement.**
>
> The performance improvement brought by the shared codebook mainly comes from its ability to better learn consistent representations across multiple views, allowing different views to be naturally aligned within a discrete latent embedding space. The discrete representation space effectively reduces noise and redundant components in continuous features, thereby highlighting the core semantic structures shared across views. In addition, the discrete codebook serves as an “information bottleneck” to some extent, encouraging the model to compress high-dimensional multi-view inputs into a set of reusable prototype vectors, which enhances the robustness and generalization ability of the learned representations.
>
> A related intuitive explanation is provided in the revised manuscript on page 4, lines 207–210, to improve the interpretability of the method design.
>
> **Whether the shared codebook design may cause semantic compression or information loss given the potentially large feature distribution heterogeneity across different views.**
>
> Specifically, in SCSD the shared codebook only functions during the quantization stage after each view is mapped into a unified latent space, while the view-specific information is already modeled independently by each view’s encoder before quantization. Therefore, the shared codebook mainly captures the semantic components shared across views and reduces redundant features that commonly exist in multi-view data. Moreover, we have view-specific encoder and decoder pathways in the model architecture, which helps mitigate the risk of semantic compression or information loss.

---

### Official Review · Reviewer_Jx7D · 2025-10-31

**Soundness:** 3
**Presentation:** 3
**Contribution:** 3
**Rating:** 6
**Confidence:** 4

**Summary:**

This paper addresses the dual-missing problem in multi-view multi-label learning by introducing the SCSD model. The proposed framework leverages a shared codebook to learn consistent representations across views, employs an adaptive weighted fusion mechanism to aggregate multi-view predictions based on their reliability, and adopts a self-distillation strategy to enhance model generalization. Experimental results on five benchmark datasets consistently demonstrate that SCSD outperforms existing state-of-the-art methods under various incomplete-view settings.

**Strengths:**

1.	The proposed weighted fusion strategy, grounded in the label correlation structure, effectively assesses the contribution of each view without introducing extra parameters, thus maintaining model efficiency.
2.	The model exhibits robust and stable performance under different levels of missing data, indicating strong adaptability to scenarios with both missing views and missing labels.
3.	The paper is clearly presented and well-structured, with coherent notation and logical flow, making it accessible and easy to follow.

**Weaknesses:**

1.	The parameter sensitivity analysis lacks a systematic evaluation of how varying the codebook size and embedding dimension affect model performance—two crucial factors that directly influence representational capacity.
2.	The problem formulation in Section 2.1 is somewhat lengthy, and the accompanying notations and descriptions could be further streamlined to improve clarity and readability.
3.	The hyperparameter tuning ranges for $\alpha$ and $\lambda$ are provided ($[1e^{-2}, 2e^{1}]$ and $[1e^{-3}, 5e^{-1}]$, respectively), but the rationale behind these specific search intervals is not discussed.
4.	It remains unclear whether the shared codebook mechanism introduces constraints related to the number of views or the feature dimensionality, and a discussion on this point would strengthen the paper’s technical completeness.

**Questions:**

Refer to Weaknesses

---

> ### Author Response · Authors · 2025-11-21
> **Response to Reviewer Jx7D Part I**
>
> We thank the reviewer for the careful evaluation of our manuscript and the valuable suggestions, which provide important guidance for improving the quality of the paper.
>
> **The influence of different codebook sizes and embedding dimensions on model performance.**
>
> We further conduct experiments to examine how the codebook size and embedding dimension influence the model performance. Specifically, we retrain and evaluate the model under multiple parameter configurations to systematically analyze how these two factors affect the representation capacity and the final performance. The results show that appropriately increasing the codebook size improves the model performance within a certain range, while overly large values lead to higher computational cost with diminishing returns. At the same time, a smaller embedding dimension helps produce more compact and more stable quantized representations, and therefore leads to better model performance.
>
> The related results and analyses are presented in detail through bar charts in the experimental section of the revised manuscript. Below, we provide a preview table of the parameter settings and the corresponding changes in model performance, where the metric reported is AP.
>
> | Corel5k                          |       |       |       |       |       |       |
> | -------------------------------- | ----- | ----- | ----- | ----- | ----- | ----- |
> | $k \downarrow / d_c \rightarrow$ | 2     | 4     | 8     | 16    | 32    | 64    |
> | 256                              | 0.445 | 0.445 | 0.438 | 0.428 | 0.414 | 0.402 |
> | 512                              | 0.445 | 0.445 | 0.439 | 0.431 | 0.417 | 0.406 |
> | 768                              | 0.446 | 0.446 | 0.441 | 0.431 | 0.421 | 0.411 |
> | 1024                             | 0.443 | 0.448 | 0.443 | 0.433 | 0.421 | 0.411 |
> | 2048                             | 0.444 | 0.447 | 0.442 | 0.432 | 0.423 | 0.414 |
> | 4096                             | 0.445 | 0.447 | 0.441 | 0.432 | 0.426 | 0.415 |
>
> | Pascal07                         |       |       |       |       |       |       |
> | -------------------------------- | ----- | ----- | ----- | ----- | ----- | ----- |
> | $k \downarrow / d_c \rightarrow$ | 2     | 4     | 8     | 16    | 32    | 64    |
> | 256                              | 0.575 | 0.576 | 0.575 | 0.571 | 0.563 | 0.554 |
> | 512                              | 0.575 | 0.576 | 0.577 | 0.569 | 0.566 | 0.556 |
> | 768                              | 0.574 | 0.578 | 0.574 | 0.572 | 0.564 | 0.556 |
> | 1024                             | 0.573 | 0.578 | 0.577 | 0.571 | 0.564 | 0.558 |
> | 2048                             | 0.577 | 0.578 | 0.576 | 0.573 | 0.566 | 0.558 |
> | 4096                             | 0.574 | 0.575 | 0.576 | 0.572 | 0.568 | 0.556 |
>
> **The writing in the problem definition section is too lengthy.**
>
> In the revised manuscript, we simplify the semantic expressions in this section and refine the related symbols and descriptions to make the presentation clearer and more readable. These revisions appear in Section 2.1 of the updated manuscript.
>
> Specifically, we appropriately compress the wording of this part by removing redundant adjectives, merging the sentences describing the missing-label matrix, simplifying the description of missing-views, and integrating the explanation of the missing-data filling strategy into the relevant paragraph. This makes the overall presentation more concise and fluent while keeping all symbol definitions and core meanings unchanged.

---

> ### Author Response · Authors · 2025-11-21
> **Response to Reviewer Jx7D Part II**
>
> **The choice of search ranges for the hyperparameters $\alpha$ and $\lambda$.**
>
> The hyperparameter $\alpha$ controls the weight of the cross-view reconstruction loss in the overall objective (see Eq. (10)). Its value needs to balance the benefit of the reconstruction term for representation learning and the need to avoid an excessively strong influence on the optimization process. Therefore, we follow common settings in related literature and consider the stable range observed in our preliminary experiments, and we set the search interval of $\alpha$ to $[1\mathrm{e}{-2}, 2\mathrm{e}{1}]$ to cover reconstruction strengths from weak to strong.
>
> The hyperparameter $\lambda$ represents the imitation weight, which balances the view-specific classifier’s alignment of the teacher distribution and its learning from ground-truth labels, as shown in Eq. (9). Since $\lambda$ forms a convex combination between these two components, its value must satisfy $0 \le \lambda \le 1$. Preliminary experiments show that the model is relatively sensitive to $\lambda$ within a small range, so within this valid interval, we further restrict the search range to $[1\mathrm{e}{-3}, 5\mathrm{e}{-1}]$ to cover effective distillation strengths from weak to moderate.
>
> **Whether the shared codebook mechanism imposes constraints on the number of views or the feature dimensions.**
>
> We appreciate the reviewer’s comment. Specifically, the shared codebook performs quantization learning in a unified latent space, and its size is determined by model design rather than by the number of views or the dimensionality of the original features. Before quantization, each view is mapped into the same latent representation space through its own encoder, which removes the impact of dimensional differences across views. Therefore, the shared codebook mechanism does not introduce additional structural constraints regarding the number of views or the feature dimensions.

---

### Author Response · Authors · 2025-11-21
**General Response to Reviewers**

We sincerely thank all reviewers for their valuable comments and for recognizing the novelty and completeness of our work. The reviewers’ feedback is highly meaningful for improving the quality of the paper. We revise the manuscript accordingly and describe the changes in our responses.

Below are the additions and modifications we make based on the reviewers’ suggestions:

- We adjust the problem definition in Section 2.1 and refine several expressions to improve clarity and readability.
- We add a description of the overall objective function $\mathcal{L}$ in Section 2.5.
- We include an analysis of the time complexity of the SCSD model at the end of Section 2.5.
- We expand the analysis of the comparative experimental results in Section 3.4, adding multiple perspectives to better explain the performance of the SCSD model.
- We add experiments evaluating different codebook sizes and embedding dimensions, including results on both the Corel5k and Pascal07 datasets (see Figures 3c and 3d), and we provide corresponding analysis at the end of Section 3.5.
- We adjust the font size in all subfigures of Figure 3 and re-optimize the overall layout to improve clarity and readability.
- We add a detailed analysis of the limitations of the SCSD model in the conclusion section (Section 4).
- We add an additional explanation of the computation in the codebook utilization analysis at the end of Appendix A.3.

---

### Meta-Review · Area_Chair_98m3 · 2026-01-05

**Summary:**

The reviewers generally agreed that this paper addressed an important and under-studied issue - dual-deletion multi-view multi-label learning - and proposed a coherent, technically sound, and motivationally reasonable framework. The shared code repository design, adaptive view weighted fusion, and self-distillation strategy were recognized, and their effectiveness was confirmed through consistent experimental results in multiple benchmark tests. Although some reviewers initially expressed concerns about parameter sensitivity, loss design principles, computational complexity, and the depth of experimental analysis, these issues were largely and convincingly resolved in the rebuttal letter and the revised manuscript through additional experiments, clearer explanations, and explicit discussions of limitations. Although a few reviewers remained cautious about actual efficiency, the remaining concerns were relatively minor and did not affect the validity or contribution of the proposed method. Overall, this paper met the acceptance criteria and is worthy of inclusion in the conference agenda.

**Reviewer Concerns:**

Addressed concerns: Most of the substantive concerns raised by the reviewers were adequately addressed in the rebuttal and revised manuscript. In particular, the authors provided systematic parameter sensitivity analyses for the shared codebook size and embedding dimension, clarified the rationale behind hyperparameter search ranges and loss-weighting design, and added a time complexity analysis to improve transparency regarding computational efficiency. The authors also expanded the experimental discussion with deeper interpretation across datasets and missingness settings, improved figure readability, and included an explicit discussion of model limitations. These additions significantly strengthen the technical completeness and clarity of the paper.

Outstanding concerns: Some concerns remain partially open. While the intuition and empirical justification for the shared codebook and discretization mechanism have been strengthened in the revision, a more formal theoretical analysis could further enhance the understanding of why discretization benefits representation consistency. In addition, although computational complexity is analyzed, empirical runtime or memory comparisons with baselines remain limited. These remaining issues mainly concern the depth of analysis and presentation, rather than the soundness of the proposed method, and do not undermine the paper’s core contributions.

**Reviewer Scores:**

Overall, the rebuttal addresses a substantial portion of the reviewers’ concerns by providing additional analyses, clearer justifications of design choices, and improved experimental discussion. Reviewer Jx7D and Reviewer 3pou would likely maintain their original scores, with a possibility of a slight increase, as the added sensitivity analysis and expanded experiments improve clarity but do not fundamentally change the contribution. Reviewer Z7Gz, who was already positive, would maintain a high score, with the rebuttal reinforcing confidence in the technical soundness of the work. Reviewer gK1A is plausibly persuaded to raise the score modestly, as the newly added complexity analysis and clarifications address several previously noted weaknesses. Some concerns related to methodological complexity and practical implications remain, reflecting broader limitations rather than clear flaws. Taken together, the rebuttal leads to a slightly more favorable but still cautious overall score distribution.

---

### Decision · Program_Chairs · 2026-01-26

Accept (Poster)